# Feasibility and Safety of Stentless Uretero-Intestinal Anastomosis in Radical Cystectomy with Ileal Orthotopic Neobladder

**DOI:** 10.3390/jcm10225372

**Published:** 2021-11-18

**Authors:** Chung Un Lee, Jong Hoon Lee, Dong Hyeon Lee, Wan Song

**Affiliations:** 1Department of Urology, Samsung Medical Center, Sungkyunkwan University School of Medicine, Seoul 06351, Korea; iatronices@naver.com (C.U.L.); smc8160921@gmail.com (J.H.L.); 2Department of Urology, Ewha Womans University Medical Center, Ewha Womans University School of Medicine, Seoul 07985, Korea; leedohn@ewha.ac.kr

**Keywords:** complications, neobladder, radical cystectomy, stentless, uretero-intestinal anastomosis

## Abstract

Background: We evaluated the feasibility and safety of stentless uretero-intestinal anastomosis (UIA) during radical cystectomy (RC) with an ileal orthotopic neobladder. Methods: We retrospectively reviewed 403 patients who underwent RC for bladder cancer between August 2014 and December 2018. The primary objective was to study the effect of stentless UIA on uretero-intestinal anastomosis stricture (UIAS), and the secondary objective was to evaluate the association between stentless UIA and other complications, including paralytic ileus, febrile urinary tract infection (UTI), and urine leakage. Kaplan–Meier survival analysis was used to estimate UIAS-free survival, and Cox proportional hazard models were applied to identify factors associated with the risk of UIAS. Results: Among 403 patients with 790 renal units, UIAS was identified in 39 (9.7%) patients and 53 (6.7%) renal units. Forty-four (83.0%) patients with UIAS were diagnosed within 6 months. The 1- and 2-year overall UIAS-free rates were 93.9% and 92.7%, respectively. Paralytic ileus was identified in 105 (26.1%) patients and resolved with supportive treatment. Febrile UTI occurred in 57 patients (14.1%). However, there was no leak of the UIA. Conclusions: Stentless UIA during RC with an ileal orthotopic neobladder is a feasible and safe surgical option. Further prospective randomized trials are required to determine the clinical usefulness of stentless UIA during RC.

## 1. Introduction

Radical cystectomy (RC) with pelvic lymph node dissection is the gold-standard treatment for muscle-invasive bladder cancer (MIBC) or high-risk, recurrent non-muscle-invasive bladder cancer (NMIBC) [1,2]. In addition, the ileal orthotopic neobladder (IONB) is currently the preferred urinary diversion (UD), as it leads to improved quality of life compared to other types of UD [3,4]. However, several complications, such as deterioration of renal function, urinary tract infection (UTI), and lithiasis, are frequently encountered [5,6] and are often associated with the obstruction of the upper urinary tract. Uretero-intestinal anastomosis stricture (UIAS) is the most common cause of upper urinary tract obstruction, and recent studies have reported that the incidence of UIAS ranges from 2.5% to 11.5% after UD following RC [6,7,8,9,10,11,12]. Ischemia and/or inflammation during ureter handling and scar formation at the anastomosis are known to increase the risk of UIAS [13]. 

Therefore, during uretero-intestinal anastomosis (UIA), intraoperative ureteral stent insertion is frequently performed for better alignment and mechanical support of the anastomosis, which might help to reduce the chance of urine leak and ureteral stricture [14]. However, the evidence for the benefits of intraoperative ureteral stent insertion is insufficient; several studies have reported no correlation between ureteral stent insertion and UIAS [15,16,17]. Moreover, other studies have suggested that ureteral stent insertion increases the risk of infectious complications by acting as a source of infection [18,19].

Understanding the role of intraoperative ureteral stent insertion during UIA will help in the management of postoperative complications. The authors claimed that UIA without intraoperative ureteral stent insertion had no effect on complications, especially in terms of UIAS. Therefore, in this study, we aimed to analyze a single surgeon’s experience with 437 consecutive patients to evaluate the feasibility and safety of stentless UIA during RC with an IONB. The primary objective of this study was to assess the effect of stentless UIA on UIAS rates, and the secondary objective was to evaluate the association between stentless UIA and other complications, including paralytic ileus, febrile UTI, and urine leakage. 

## 2. Materials and Methods

### 2.1. Study Population

This study was approved by the Institutional Review Board of Ewha Womans University Mokdong Hospital (IRB No. 2019-02-004), and the IRB waived the requirement for informed consent due to the retrospective nature of this study. All study protocols were performed in accordance with the principles of the Declaration of Helsinki. We retrospectively reviewed a prospectively maintained database of 437 patients who underwent RC for MIBC or high-risk recurrent NMIBC performed by a single urologic oncology surgeon between August 2014 and December 2018. We excluded 37 patients who underwent ileal conduit UD after RC. Finally, 403 patients who underwent Studer IONB creation after RC were analyzed in this study. Among these patients, 16 had previously undergone radical nephrectomy (*n* = 8) or radical nephroureterectomy (*n* = 8), and 790 renal units were analyzed. 

### 2.2. Data Collection

The medical records of all patients at the time of surgery were reviewed, and their clinical and pathological characteristics were evaluated, including age at surgery; sex; body mass index (BMI); comorbidities, such as diabetes mellitus (DM); the American Society of Anesthesiologists (ASA) score; preoperative and postoperative radiation and/or chemotherapy; pathologic T and N stage; surgical margin status, operation time; estimated blood loss; and hospital stay.

UIAS was defined as hydronephrosis and/or acute increase in serum creatinine level with decreased urine output with radiologic evidence of obstruction at the level of UIA that required percutaneous nephrostomy tube placement or ureteral stent placement with/without endoscopic management. Paralytic ileus was defined as radiologic findings and accompanying symptoms, such as continued fasting, nasogastric tube insertion, or discontinuation of oral intake. Febrile UTI was defined as bacteriuria with a fever of 38 degrees or higher that required antibiotic treatment. Urine leakage was defined as sign of contrast leakage at uretero-intestinal anastomosis site from cystogram or computed tomography (CT). UIAS-free survival was calculated from the date of UD to the date of diagnosis of UIAS or the final follow-up date on which the patient was without UIAS.

### 2.3. Surgical Technique of UIA

The standard procedure for RC, including standard lymphadenectomy, was conducted as an open technique in all patients and performed by an experienced urologic surgeon. In general, RC included removal of the prostate and seminal vesicles in men and removal of the ovaries and uterus in women [20]. All UIAs were performed using the Bricker technique, emphasizing a separate insertion, tension-free and widely spatulated anastomosis, end-to-side refluxing fashion, with interrupted sutures using a 4-0 absorbable polyglactin suture [21]. The notable parts of our techniques are as follows: first, when performing interrupted anastomosis, sutures are initially placed at the 6 o’clock and 12 o’clock positions. Second, the ureter’s mucosa is cut parallel to the bowel mucosa, which helps maintain watertight sutures. Third, tension is applied or released to each mucosa to prevent gaps formation during suturing. Finally, ureteral stents are not placed, and a leakage test is performed to check for watertightness after the suturing is completed. Additionally, when skeletonizing ureters, periureteral tissue is contained as much as possible for the recovery of vascularity. Additionally, repositioning of the neobladder is performed by counterclockwise rotation, allowing the proximal tubular segment to pass from left to right, wrapping around the neobladder.

### 2.4. Perioperative Management

All patients took 1 gallon of polyethylene glycol-electrolyte solution and low-residue diet/clear liquid diet the day before surgery for preoperative bowel preparation. We applied a standardized postoperative care protocol and enhanced recovery after surgery (ERAS) protocol was applied since 2018. Ambulation began on postoperative day (POD) one or sooner depending on the patients’ condition. Diet started on POD four or five with oral water ingestion, and a liquid diet and soft diet were provided consecutively as tolerated. Abdominal x-ray was performed daily from POD one until the day of soft diet intake. Abdomen-pelvis CT was obtained when paralytic ileus persisted. Jackson Pratt drain was removed POD seven, and 24F Foley catheter was removed POD 14 depending on cystogram result.

### 2.5. Patient Follow-up Protocol 

Each patient was followed up according to general recommendations and institutional regulations. The follow-up protocol for patients after RC included an initial visit 2–4 weeks after discharge, then quarterly for the first 2 years, semiannually for the next 3 years, and annually thereafter. Patients underwent laboratory tests, urine analysis, and routine imaging with CT of the chest, abdomen, and pelvis at every visit. When renal function deterioration in laboratory tests and/or hydronephrosis was identified on CT, patients underwent further evaluation, typically with diuretic renogram and/or antegrade pyelography after percutaneous nephrostomy. 

### 2.6. Statistical Analysis

Descriptive statistics were used to characterize the entire cohort. Continuous data are presented as median (range) or mean (standard deviation (SD)), and categorical data are presented as absolute value (percentage). An independent *t*-test was used to compare continuous variables, and Pearson’s chi-square test or Fisher’s exact test was used to compare categorical variables. Kaplan–Meier survival analysis was used to estimate UIAS-free survival, and differences were stratified using the log-rank test. All statistical analyses were performed using IBM SPSS Statistics for Windows, version 23.0 (IBM Corp. Armonk, NY, USA). Two-sided *p*-values < 0.05 were considered statistically significant.

## 3. Results

The clinicopathologic characteristics of the 403 patients who underwent the Studer IONB creation after RC are summarized in Table 1. In the entire cohort, the median (range) age at RC was 66.0 (27.0–84.0) years. The male-to-female ratio was approximately 5:1. UIAS was confirmed in 39 (9.7%) of the 403 patients and 53 (6.7%) of the 790 renal units. When patients were categorized according to UIAS, the groups did not differ significantly in terms of age, sex, BMI, the ASA score, prior radiotherapy, adjuvant chemotherapy, or final pathologic findings. Intraoperative findings did not differ significantly. However, a history of DM and administration of neoadjuvant chemotherapy were significantly higher (*p* = 0.017 and *p* < 0.001, respectively) in the UIAS group.

Figure 1 shows the overall UIAS-free rates estimated using the Kaplan–Meier method. Forty-four (83.0%) patients with UIAS were diagnosed within six months. The one- and two-year overall UIAS-free rates were 93.9% and 92.7%, respectively. The latest diagnosis of UIAS occurred 23 months after RC. There was no significant difference in UIAS according to laterality (*p* = 0.922, Figure 2).

The outcomes of Cox proportional hazard regression analysis to identify the association between clinicopathologic characteristics and development of UIAS are presented in Table 2. On univariate analysis, DM (hazard ratio (HR) = 2.258; 95% confidence interval (CI): 1.144–4.458; *p* = 0.019), neoadjuvant chemotherapy (HR = 3.395; 95% CI: 1.720–6.703; *p* < 0.001), and operation time (HR = 1.006; 95% CI: 1.000–1.012; *p* = 0.049) were found to be significantly associated with UIAS. Multivariable analyses revealed that a history of DM (HR = 2.564; 95% CI: 1.287–5.109; *p* = 0.007) and neoadjuvant chemotherapy (HR = 3.432; 95% CI: 1.713–6.876; *p* = 0.001) were independently associated with a significantly increased risk of UIAS.

Paralytic ileus was identified in 105 (26.1%) patients and resolved with conservative treatment within a week. Febrile UTI occurred in 57 (14.1%) patients and was treated with third generation cephalosporine antibiotics. Urine leakage along the neobladder suture line developed in 27 (6.7%) patients due to an obstructed Foley catheter, but there was no leakage from the UIA.

## 4. Discussion

In this study, among 403 patients with 790 renal units who underwent RC with IONB creation for bladder cancer, UIAS was identified in 39 (9.7%) patients and 53 (6.7%) renal units. These results are comparable to those of previous studies in which an intraoperative ureteral stent was inserted during UIA. Paralytic ileus and febrile UTI were identified in 108 (26.8%) and 58 (14.4%) patients, respectively. However, there was no urine leakage from the UIA. These results are significant in that they provide evidence supporting UIA without an intraoperative ureteral stent. To the best of our knowledge, this is the first study to investigate the feasibility and safety of stentless UIA after RC with IONB creation in a large cohort. 

To date, several studies have reported the outcomes of UIAS with intraoperative ureteral stent during UD following RC, and these results are summarized in Table 2 [6,7,8,9,10,11,12]. They reported that the incidence of UISA ranged from 2.5% to 11.5% depending on patients’ characteristics and surgical technique. Our results of UIAS in 39 (9.7%) patients and 53 (6.7%) renal units were comparable to those of previous studies, supporting the feasibility of stentless UIA.

Furthermore, among the utilized surgical techniques, the type of suture in UIA has also been reported as a particularly important factor in the development of UIAS. Large et al. studied the effect of suture type on the UIAS rate and reported that running anastomosis (12.7%) was significantly associated with UIAS than interrupted anastomosis (8.5%) (HR 1.92, 95% CI 1.00–3.70, *p* = 0.05) [13]. In our study, all UIA procedures were performed with interrupted anastomosis and showed a comparable UIAS rate (6.7%). We also tried to minimize the gap between the ureter mucosa and bowel mucosa by adjusting the tension during interrupted anastomosis. 

As UIAS is considered a late complication, there is a variation in reporting the timing of the development of UIAS. Previous studies have reported that the median time from surgery to diagnosis of UIAS ranged from seven to nine months [21,22], and three-quarters of cases of UIAS were identified within one year [22]. However, in one study, UIAS was reported as late as 160 months after RC [8]. In our study, 44 (83.0%) cases of UIAS were diagnosed within six months, and the latest diagnosed UIAS was 23 months after RC. The median follow-up period was 24.3 months, which is considered sufficient to cover most cases of UIAS. 

Regarding the laterality of UIAS, previous studies have reported that UIAS occurred more frequently on the left side, as it passed under the sigmoid mesentery, thus increasing the angulation and having a longer course [13,21,22,23]. However, in our study, there was no significant difference in the occurrence of UIAS according to laterality. A possible explanation for this discrepancy is that, when dissecting the left ureter, electrocauterization was minimized, periureteral tissue was contained as much as possible, and sufficient space was made when passing beneath the sigmoid mesentery. In addition, the incidence of bilateral UIAS was higher in this study than in previous studies [9,10,12], and further studies on the correlation with stentless UIA are needed.

To date, several studies have reported postoperative complications after RC, and these results are summarized in Table 3 [24,25,26,27,28]. Studies have reported that the rate of paralytic ileus ranged from 7.1% to 22.2%, which is slightly lower than the rate reported in our study (26.1%). However, symptoms of paralytic ileus are observer-dependent clinical features, and definitions of paralytic ileus varied for each study. In our study, paralytic ileus was widely defined as discontinuation of oral intake while feeding as well as continued pasting due to the absence of bowel movement. In addition, there was no urine leakage from the UIA, and all leakages were from the neobladder suture line. During cystography, contrast refluxed to the renal pelvis and leakage of contrast was not observed from the UIA. We believe that precise handing of the ureter and meticulous suture results in patent and watertight anastomosis at the UIA.

In the present study, febrile UTI was identified in 57 patients (14.1%). Donat et al. reported that infectious complications were lower in the non-intraoperative stent group than in the intraoperative stent group (14% vs. 32%, *p* = 0.004) [19], and the risk of bacteremia and UTI increased approximately one day after stent removal [18]. Possible explanations for the increase in UTI after stent removal include bacterial colonization, stent obstruction from mucus, and urinary reflux caused by the stent. Collectively, stentless UIA might be a contributing factor that can reduce postoperative infectious complications.

Overall, our study showed comparable UIAS rate and excellence in terms of febrile UTI and urine leakage at UIA but somewhat poor results in paralytic ileus compared with studies of intraoperative ureteral stent insertion. Intraoperative ureteral stent insertion was performed conventionally, and it is considered natural. However, our study showed the feasibility and safety of stentless UIA during RC with an IONB and suggested the introduction of stentless procedure.

Despite its potential clinical implications, this study has some limitations. First, this was a retrospective study conducted by a single surgeon at a tertiary referral center, which raises concerns regarding selection bias. Additionally, we cannot compare directly between intraoperative ureteral stent insertion group and stentless group. Therefore, a better designed study with larger amount of data is needed, and through this study, independent predictors of UIAS can be found. Furthermore, as all procedures were performed in a high-volume center specialized for IONB creation after RC, and it might result in low rates of complications. It is prudent to apply the surgical techniques in a center with less experience. Second, as all surgical procedures were performed using an open technique, additional studies are required to validate the outcomes of stentless UIA in robotic surgery or conduit diversion. Finally, the length of the stricture, which could affect the treatment outcomes of UIAS, was not measured. Therefore, the characteristics of UIAS that occurred during stentless UIA have not been sufficiently analyzed.

## 5. Conclusions

Stentless UIA during RC with an IONB is a feasible and safe surgical option. The UIAS is usually present a few months after RC with an IONB, and the incidence of UIAS is comparable to that in studies with intraoperative ureteral stent insertion. A history of DM and neoadjuvant chemotherapy were significantly associated with an increased risk of UIAS. Further prospective randomized trials are required to achieve the clinical usefulness of stentless UIA during RC with an IONB.

## Figures and Tables

**Figure 1 jcm-10-05372-f001:**
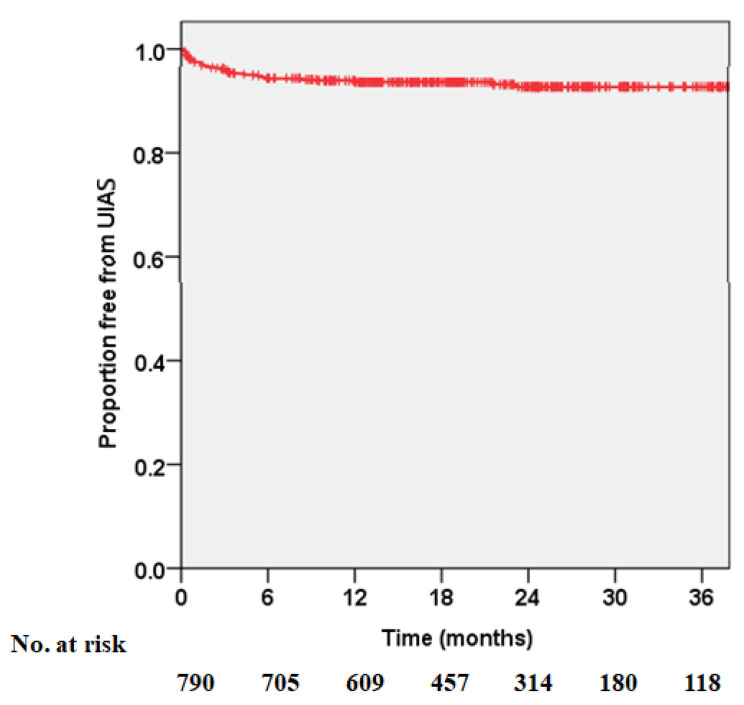
Kaplan–Meier curve showing the overall uretero-intestinal anastomosis stricture-free rate after radical cystectomy with ileal orthotopic neobladder.

**Figure 2 jcm-10-05372-f002:**
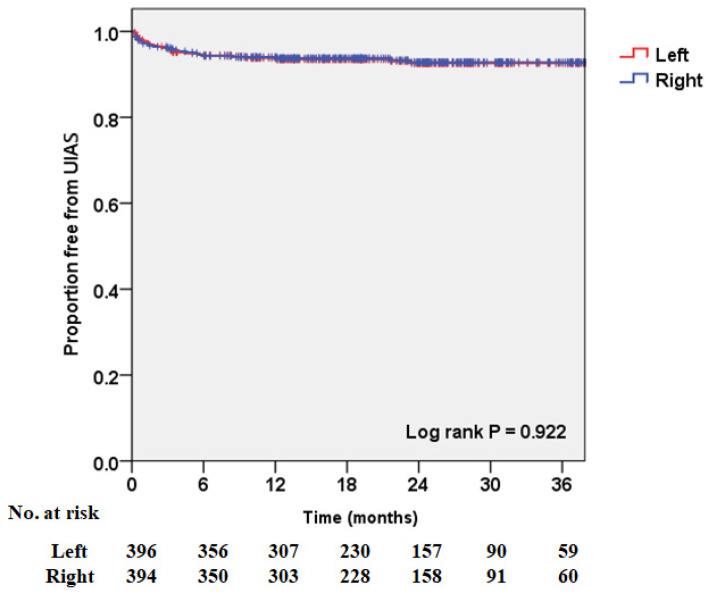
Kaplan–Meier curve showing the uretero-intestinal anastomosis stricture-free rate after radical cystectomy with ileal orthotopic neobladder according to laterality.

**Table 1 jcm-10-05372-t001:** Baseline characteristics of 403 patients who underwent radical cystectomy with ileal orthotopic neobladder.

Variable	Total	Uretero-Intestinal Anastomosis Stricture		*p*
		Yes	No	
No. of patients	403 (100.0)	39 (9.7)	364 (90.3)	
Age at surgery, years				0.622
Median (range)	66.0 (27.0–84.0)	67.0 (34.0–81.0)	66.0 (27.0–84.0)	
Mean (SD)	63.9 (10.3)	64.7 (10.2)	63.8 (10.4)	
Sex, N (%)				0.493
Male	336 (83.4)	31 (79.5)	305 (83.8)	
Female	67 (16.6)	8 (20.5)	59 (16.2)	
Body mass index, kg/m^2^				0.068
Median (range)	24.0 (15.0–38.7)	25.3 (17.3–31.6)	23.9 (15.0–38.7)	
Mean (SD)	24.2 (3.2)	25.1 (3.4)	24.1 (3.2)	
Diabetes mellitus, N (%)				0.017
Yes	69 (17.1)	12 (30.8)	57 (15.7)	
No	334 (82.9)	27 (69.2)	307 (84.3)	
ASA score, N (%)				0.119
1	53 (13.2)	2 (5.1)	51 (14.0)	
2–3	350 (86.8)	37 (94.9)	313 (86.0)	
Neoadjuvant chemotherapy, n (%)				< 0.001
Yes	48 (11.9)	12 (30.8)	36 (9.9)	
No	355 (88.1)	27 (69.2)	328 (90.1)	
Preoperative radiotherapy, n (%)				0.400
Yes	5 (1.2)	1 (2.6)	4 (1.1)	
No	398 (98.8)	38 (97.4)	360 (98.9)	
Operation time, min				0.051
Median (range)	265.0 (175.0–480.0)	275.0 (195.0–455.0)	265.0 (175.0–480.0)	
Mean (SD)	271.4 (46.9)	285.4 (55.2)	269.9 (45.8)	
Estimated blood loss, mL				0.225
Median (range)	500.0 (100.0–2200.0)	500.0 (200.0–1500.0)	475.0 (100.0–2200.0)	
Mean (SD)	520.9 (279.2)	572.6 (311.9)	515.4 (275.4)	
Pathologic T stage, N (%)				0.102
≤pT2 (organ confined)	255 (63.3)	20 (51.3)	235 (64.6)	
≥pT3 (locally advanced)	148 (36.7)	19 (48.7)	129 (35.4)	
Pathologic N stage, N (%)				0.344
N0/Nx	304 (75.4)	27 (69.2)	277 (76.1)	
N1–3	99 (24.6)	12 (30.8)	87 (23.9)	
Surgical margin status, N (%)				0.196
Positive	64 (15.9)	9 (23.1)	55 (15.1)	
Negative	339 (84.1)	30 (76.9)	309 (84.9)	
Hospital stay, days				0.287
Median (range)	15.0 (6.0–51.0)	16.0 (13.0–36.0)	15.0 (6.0–51.0)	
Mean (SD)	17.7 (6.8)	18.8 (5.9)	17.6 (6.9)	
Adjuvant chemotherapy, n (%)				0.717
Yes	175 (43.4)	18 (46.2)	157 (43.1)	
No	228 (56.6)	21 (53.8)	207 (56.9)	
Renal units, n (%)				
Bilateral	387 (96.0)			
Left only	9 (2.2)			
Right only	7 (1.8)			
Laterality of stricture, n (%)				
Left only		13 (33.3)		
Right only		12 (30.8)		
Bilateral		14 (35.9)		
Follow-up, months				0.329
Median (range)	24.3 (2.9–56.7)	24.4 (12.2–29.0)	24.0 (2.9–56.7)	
Mean (SD)	24.4 (11.3)	23.6 (3.7)	24.5 (11.8)	

ASA, American Society of Anesthesiologists; SD, standard deviation.

**Table 2 jcm-10-05372-t002:** Comparison of uretero-intestinal anastomosis stricture between current study and large cohort studies with intraoperative ureteral stent indwelling during radical cystectomy.

Study	Amin et al. [12]	Goh et al. [11]	Yang et al. [10]	Shah et al. [9]	Shimko et al. [6]	Hautmann et al. [8]	Shabsigh et al. [7]	Current Study
Study interval	1995–2014	2009–2014	1980–2008	1971–2008	1980–1998	1986–2008	1995–2005	2014–2018
No. of patients	2888	1449	2285	1964	1057	923	1142	403
Male, %	74.1	79.9	81.4	NA	79.7	86.1	75.5	83.4
Age, year								
Median (range)	68 (60–75) *	NA	68 (62–75) *	NA	69 (31–92)	NA	68 (60–75) *	66 (27–84)
BMI, kg/m^2^								
Median (range)	28 (25–30) *	NA	27.0(24–30) *	NA	NA	NA	27.1(24–30) *	24.0(15.0–38.7)
Neobladder, %	33.6	7.9	21.6	80.2	0	100	36.6	100
UIAS								
Renal units, n (%)	NA	NA	NA	51 (NA)	NA	NA	NA	53 (6.7)
Patients, %	4.3	4.2–8.3	8.4	2.5	11.5	11.1	3.9	9.7
Male, %	NA	NA	82.3	85.7	NA	NA	NA	79.5
Laterality, %								
Left only	53.7	NA	53.1	66.0	NA	NA	NA	33.3
Right only	40.7	NA	28.1	29.0	NA	NA	NA	30.8
Bilateral	5.7	NA	18.8	5.0	NA	NA	NA	35.9

* IQR; BMI, body mass index; UIAS, uretero-intestinal anastomosis stricture; IQR, interquartile range; NA, not available.

**Table 3 jcm-10-05372-t003:** Comparison of complications between current study and recent studies with intraoperative ureteral stent indwelling during radical cystectomy.

Study	Malangone-Monaco et al. [24] (2020)	Vetterlein et al. [26] (2020)	Haider et al. [25] (2019)	Parekh et al. [27] (2018)	Hirobe et al. [28] (2018)	Current Study
Study design	Retrospective	Retrospective	Retrospective	Multicenter randomized controlled trial	Prospective	Retrospective
Study interval	2005–2015	2009–2017	2009–2015	2011–2014	2010–2014	2014–2018
No. of patients	4648	506	217	152	185	403
Male, %	78.7	79.0	78.3	84.0	79.4	83.4
Age, year						
Median (range)	67.6 (NA)	69 (62–74) *	72 (64–78.5) *	67 (37–85)	72 (39–89)	66 (27–84)
BMI, kg/m^2^			26.1	28.2	23.4	24.0
Median (range)	NA	26 (24–29) *	(23.2–29.7) *	(24.9–31.7) *	(15.2–34.2)	(15.0–38.7)
Neobladder, %	NA	27	28.1	20.0	8.1	100
Complications, %						
Paralytic ileus	15.8	7.1	NA	20.0	22.2	26.1
Febrile UTI	25.3	62.0	19.4	26.0	29.7	14.1
Urine leak at UIA	NA	2.0	NA	3.0	2.7	0

* IQR; BMI, body mass index; UTI, urinary tract infection; UIA, uretero-intestinal anastomosis; NA, not available; IQR, interquartile range.

## Data Availability

The dataset used and/or analyzed during the current study is available from the corresponding author upon reasonable request.

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
