# Peer review of "Feasibility and Safety of Stentless Uretero-Intestinal Anastomosis in Radical Cystectomy with Ileal Orthotopic Neobladder"

_jcm, 2021, doi:10.3390/jcm10225372_

Round 1

Reviewer 1 Report

Authors are congratulated for providing homogeneous and huge population of patients treated with radical cystectomy and orthotopic ileal neobladder (Studer configuration). They focused on rates of ureteral neobladder anastomosis and they want to test if stentless anastomosis provides acceptable functional results. Overall the manuscript well written and the topic is of interest for the Readers. However, I believe there are several major and minor comments that need to be addressed before acceptance:

  • Authors provide results on the feasibility of the technique, however, due to the lack of a control population (anastomosis with stent) no direct comparison between the two groups and the real advantage of the stentless technique could be assess, therefore, authors should tone down  their message and should improve their discussion section ( i.e. necessity of a case control study before clinical applicability)
  • A detailed focus on perioperative management of patient and of the devices is needed. For example, did the authors have a fast track protocol? Please add a specific section in the materials and methods and provide information about catheter removal. time from surgery to x-ray cystogram etc.
  • I'm quite surprised about the low rates of urinary leakages in such a big population, especially if compared to the other studies available today. Do the authors have an explanation for these results? (expert surgeon? Surgical technique? Other aspects?) In my opinion it is possible these low rates are due to the definition of “urinary leakage” in the current manuscript. Specifically, in my opinion it is difficult to have clinically significant leakages (urinomas) due to neobladder permeability since all these patients have a bladder catheter. Please comment.
  • Authors should comment about the low rates of strictures (events) in this population. Specifically, they cannot perform multivariable cox regression models testing association between patient/ surgical/pathological characteristics and main events (underpowered analyses). In consequence, I suggest to remove these models and only show difference in distribution between events vs no events (table 1). Moreover, this passage should be commented in the discussion (need of other studies with more events to find independent predictors of stricture).
  • Discussion could be restructured in order to underline the main findings of the current analysis and not only previously available results. Suggestions about the clinical applicability of the results is encouraged.

Author Response

1. Authors provide results on the feasibility of the technique, however, due to the lack of a control population (anastomosis with stent) no direct comparison between the two groups and the real advantage of the stentless technique could be assess, therefore, authors should tone down their message and should improve their discussion section ( i.e. necessity of a case control study before clinical applicability)

 -->The absence of a control group is an obvious limitation. However, we tried to confirm feasibility and safety of stentless anastomosis in a large number of patients. Contrary to your concerns, comparable outcomes were confirmed compared to previous studies. Additionally, it is necessary to confirm the results of this study through a prospective randomized trial. We added them at “Discussion (limitation)” section.

2. A detailed focus on perioperative management of patient and of the devices is needed. For example, did the authors have a fast track protocol? Please add a specific section in the materials and methods and provide information about catheter removal. time from surgery to x-ray cystogram etc.

--> We added “2.4 Perioperative management” at “Materials and Methods” section.

3. I'm quite surprised about the low rates of urinary leakages in such a big population, especially if compared to the other studies available today. Do the authors have an explanation for these results? (expert surgeon? Surgical technique? Other aspects?) In my opinion it is possible these low rates are due to the definition of “urinary leakage” in the current manuscript. Specifically, in my opinion it is difficult to have clinically significant leakages (urinomas) due to neobladder permeability since all these patients have a bladder catheter. Please comment.

--> The reasons for low rates of urinary leakages are below.

  First, RC was performed by an expertized surgeon at a high-volume center specialized for OINB creation after RC (100 – 150 cases per year). Second, ureter’s mucosa is cut parallel to the bowel mucosa, which helps maintain watertight sutures, and minimize the gap between the ureter mucosa and bowel mucosa by adjusting the tension during interrupted anastomosis. Third, leakage test after UIA anastomosis was performed. Finally, we perform reposition of neobladder to reduce leakage by overflow. That is through counterclockwise rotation, proximal tubular segment to pass from left to right, wrapping around the neobladder.

 As you mentioned, there is a possibility of a little amount of leakage. However, it can be absorbed in the peritoneal cavity, and it thought to be feasible if there is no clinical problem.

4. Authors should comment about the low rates of strictures (events) in this population. Specifically, they cannot perform multivariable cox regression models testing association between patient/ surgical/pathological characteristics and main events (underpowered analyses). In consequence, I suggest to remove these models and only show difference in distribution between events vs no events (table 1). Moreover, this passage should be commented in the discussion (need of other studies with more events to find independent predictors of stricture).

--> We mentioned that “In our study, all UIA procedures were performed with interrupted anastomosis and showed a comparable UIAS rate (6.7%). We also tried to minimize the gap between the ureter mucosa and bowel mucosa by adjusting the tension during interrupted anastomosis” in Discussion section.

 As your recommendation, we removed multivariable cox regression models and associated sentences. And we commented need for further study and finding independent predictors at “Discussion – limitation” section.

5. Discussion could be restructured in order to underline the main findings of the current analysis and not only previously available results. Suggestions about the clinical applicability of the results is encouraged.
--> We added a paragraph that summarized our study and other studies, and suggested clinical applicability.

Reviewer 2 Report

The topis is nice and the paper is well written. 

However some modifications are needed.

Concerning surgical technique:

  • which was the approach? Open? Laparoscopic or robotic? -
  • Was lymph node dissection always performed? Which template? In this surgical step, the ureter is skeletonized and strictures due to devascularization may be developed in the future at the leve of anastomosis. Did the author applied any precautions to avoid this complication?

Concerning post-operative complications, other than ileum, sometimes the overflow of urine through a stentless anastomosis may cause not only a urinary leakage but a real anastomosis dehiscence. It its never reported in the present experience. Therefore an untreated urinoma may also produce a dehiscence of bowel anastomosis with further major consequences.

My fear is that this technique needs more extensive explanations, discussion and comment on such favourable results.

Author Response

1. Which was the approach? Open? Laparoscopic or robotic?

--> We mentioned open technique in “Materials and Methods – 2.3 surgical technique of UIA”.

2. Was lymph node dissection always performed? Which template? In this surgical step, the ureter is skeletonized and strictures due to devascularization may be developed in the future at the level of anastomosis. Did the author applied any precautions to avoid this complication?
--> We mentioned “standard procedure for RC, including standard lymphadenectomy” in “Materials and Methods – 2.3 surgical technique of UIA”. And we added a sentence about recovery of vascularity of ureters at the last of “Materials and Methods – 2.3 surgical technique of UIA” section.

3. Concerning post-operative complications, other than ileum, sometimes the overflow of urine through a stentless anastomosis may cause not only a urinary leakage but a real anastomosis dehiscence. It is never reported in the present experience. Therefore an untreated urinoma may also produce a dehiscence of bowel anastomosis with further major consequences.

My fear is that this technique needs more extensive explanations, discussion and comment on such favourable results.

--> Intraoperative ureteral stent insertion was performed conventionally, and it is considered natural. So, we understand your concerning about stentless UIA. This study was based on data of a high-volume center specialized for OINB creation after RC (100 – 150 cases per year). There might be a concerning about overflow of urine in stentless anastomosis, however to reduce the problem, we perform reposition of neobladder. That is through counterclockwise rotation, proximal tubular segment to pass from left to right, wrapping around the neobladder. Through this procedure, we can reduce leakage at anastomosis site caused by overflow. Briefly, leakage test after UIA anastomosis and reposition of neobladder lessen the possibility of overflow, therefore possibility of severe leakage and urinoma is low. These are added at “Materials and Methods – 2.3 surgical technique of UIA” section and “Discussion – limitation” section.